# Identification of Hemagglutinin Mutations Caused by Neuraminidase Antibody Pressure

Fei Wang,[a,b] Zhimin Wan,[a,b] Yajuan Wang,[a,b] Jinsen Wu,[a,b] Hui Fu,[a,b] Wei Gao,[a,b] Hongxia Shao,[a,b,c] Kun Qian,[a,b,c] 🆔 Jianqiang Ye,[a,b,c] 🆔 Aijian Qin[a,b,c]

[a]Ministry of Education Key Laboratory for Avian Preventive Medicine, Yangzhou University, Yangzhou, Jiangsu, China
[b]Jiangsu Co-innovation Center for Prevention and Control of Important Animal Infectious Diseases and Zoonoses, Yangzhou, Jiangsu, China
[c]Joint International Research Laboratory of Agriculture and Agri-Product Safety of Ministry of Education of China, Yangzhou University, Yangzhou, Jiangsu, China

Fei Wang and Zhimin Wan contributed equally to this article. Author order was determined in order of initiating this research project.

**ABSTRACT** The balance in the functions of hemagglutinin (HA) and neuraminidase (NA) plays an important role in influenza virus genesis. However, whether and how N2 neuraminidase-specific antibodies may affect the attributes of HA remains to be investigated. In this study, we examined the presence of amino acid mutations in the HA of mutants selected by incubation with N2-specific monoclonal antibodies (MAbs) and compared the HA properties to those of the wild-type (WT) A/Chicken/Jiangsu/XXM/1999 (XXM) H9N2 virus. The higher NA inhibition (NI) ability of N2-specific MAbs was found to result in greater proportions of mutations in the HA head. The HA mutations affected the thermal stability, switched the binding preferences from $\alpha$2,6-linked sialic acid receptor to $\alpha$2,3-linked sialic acid receptor, and promoted viral growth in mouse lungs. These mutations also caused significant HA antigenic drift as they decreased hemagglutination inhibition (HI) titers. The evolutionary analysis also proved that some HA mutations were highly correlated with NA antibody pressure. Our data demonstrate that HA mutations caused by NA-specific antibodies affect HA properties and may contribute to HA evolution.

**IMPORTANCE** HA binds with the sialic acid receptor on the host cell and initiates the infection mode of influenza virus. NA cleaves the connection between receptor and HA of newborn virus at the end of viral production. The HA-NA functional balance is crucial for viral production and interspecies transmission. Here, we identified mutations in the HA head of H9N2 virus caused by NA antibody pressure. These HA mutations changed the thermal stability and switched the receptor-binding preference of the mutant virus. The HI results indicated that these mutations resulted in significant antigenic drift in mutant HA. The evolutionary analysis also shows that some mutations in HA of H9N2 virus may be caused by NA antibody pressure and may correlate with the increase in H9N2 infections in humans. Our results provide new evidence for HA-NA balance and an effect of NA antibody pressure on HA evolution.

**KEYWORDS** hemagglutinin mutations, neuraminidase antibody pressure, influenza virus, receptor binding preferences, antigenic drift

Influenza A viruses (IAVs) are important pathogens of both animals and humans. Hemagglutinin (HA) and neuraminidase (NA) are the most abundant glycoproteins on IAV. Vaccines, including inactivated or attenuated IAVs, induce neutralizing antibodies against HA and NA (1). However, changes in HA and NA can help virus escape from humoral immunity by the introduction of glycans or amino acid substitutions and deletions (2–4). Interestingly, whatever changes take place in HA and NA, the basic functions of HA and NA do not change.

Address correspondence to Jianqiang Ye, jqye@yzu.edu.cn, or Aijian Qin, aijian@yzu.edu.cn.

The authors declare no conflict of interest.

HA is a trimeric glycoprotein, and the mature HA monomer consists of disulfide-linked chains HA1 and HA2. The receptor-binding sites (RBS) in HA1 help virus attach to sialic acid receptors on host cells. Avian influenza viruses in nature prefer binding to $\alpha$2,3-linked sialic acid receptors, whereas human influenza viruses preferentially bind to $\alpha$2,6-linked sialic acid receptors (5, 6). HA2 is mainly involved in viral penetration by mediating fusion of the endosomal and viral membranes. Antigenic drift in HA, especially in the RBS, may change the HA binding properties and even influence species tropism (7, 8). NA is a tetrameric glycoprotein with a mushroom-like head (9). The function of NA is to remove sialic acid residues from viral HA and infected cells during both viral entry and release from cells (10, 11). The NA enzymatic activity contributes to the mobility of viruses before viral attachment and efficient release of progeny virions.

The functional balance of the receptor-binding HA and receptor-destroying NA is crucial for viral mobility in the airway of hosts (12). The HA-NA balance is also necessary to viral production, host adaption, and cross-species transmission (13–15). Higher binding ability of HA to receptors would increase the cleavage efficiency of NA (10). However, higher activity or inhibition of HA or NA would break the HA-NA balance and result in both HA and NA mutations (16–18). Mutations in NA are also able to drive compensatory mutations in HA of influenza virus (19). We previously mapped antigenic variations in NA of H9N2 IAVs with a panel of 22 monoclonal antibodies (MAbs) (20). In this study, we identified HA mutations in viruses selected by N2 neuraminidase-specific neutralizing MAbs. These HA mutations changed the receptor tropism, antigenic structure, and growth characteristics of H9N2 virus, which revealed the evolution of HA in the case of NA antibody pressure.

## RESULTS

**NA antibody pressure resulted in mutations in the HA head.** MAbs A2A3, A4C6, A5D12, A3C9, A6A7, and B4D6 against NA of A/Chicken/Jiangsu/XXM/1999 (XXM) virus were previously proved to have neutralizing ability in a microneutralization (MN) assay and high NA inhibition (NI) ability in an enzyme-linked lectin assay (ELLA) (20). In this research, the NI ability of these MAbs for the wild-type (WT) XXM virus was further measured in a Mu-NANA [2′-(4-methylumbelliferyl)-$\alpha$-D-N-acetylneuraminic acid] assay. In contrast to the MN assay and ELLA results, only MAbs A2A3, A3C9, and B4D6 had significant NI effects in the Mu-NANA assay, while the other MAbs and a nonneutralizing MAb, B6G5, all had very weak (<20%) or no NI effect on XXM virus (Fig. 1A). The Mu-NANA assay results indicated that MAbs A2A3, A3C9, and B4D6 could effectively block Mu-NANA, which is a small-molecule substrate, from binding with NA.

The indirect effect of the NA antibody pressure of these MAbs on the HA-NA balance was evaluated by HA sequencing. Interestingly, combined mutations at amino acid positions 166, 198, and 234 (H9 numbering) in HA were found in XXM H9N2 viruses incubated with neutralizing MAbs but not in the viruses incubated with non-neutralizing MAb B6G5 (Fig. 1B). For viruses incubated with MAbs A2A3 and A3C9, which showed higher NI activities in the Mu-NANA assay (Fig. 1A), higher percentages (>90%) of changes of codons at positions 166, 198, and 234 in HA of the selected viruses were shown in the sequencing report. The other MAbs, A4C6, A5D12, B4D6, and A6A7, showed weaker NI activities in the Mu-NANA assay but still resulted in low levels of codon changes at the same positions in HA. The nucleotide codon at position 166 changed from GAC to AAC, which resulted in a D166N mutation in HA (Fig. 1B). Similarly, the dual nucleotide changes of ACA to GTA led to a T198V mutation, and another codon change of CTG to CAG led to an L234Q mutation.

Moreover, HA mutations were also identified in 4 of 10 escape mutants purified by plaque assay (Table 1), e.g., combined amino acid mutations D166N/T198V/N201D/L234Q were found in the HA of MAb A2A3 escape mutant mA2A3-2, while escape mutant mA2A3-4 only had a single mutation, T220I. Both MAb A3C9 escape mutant mA3C9-1 and mA3C9-3 had D166N/A168T/T198V/L234Q mutations in the HA. All of these mutations took place in the HA head domain (Fig. 1C). Residues 198, 201, and

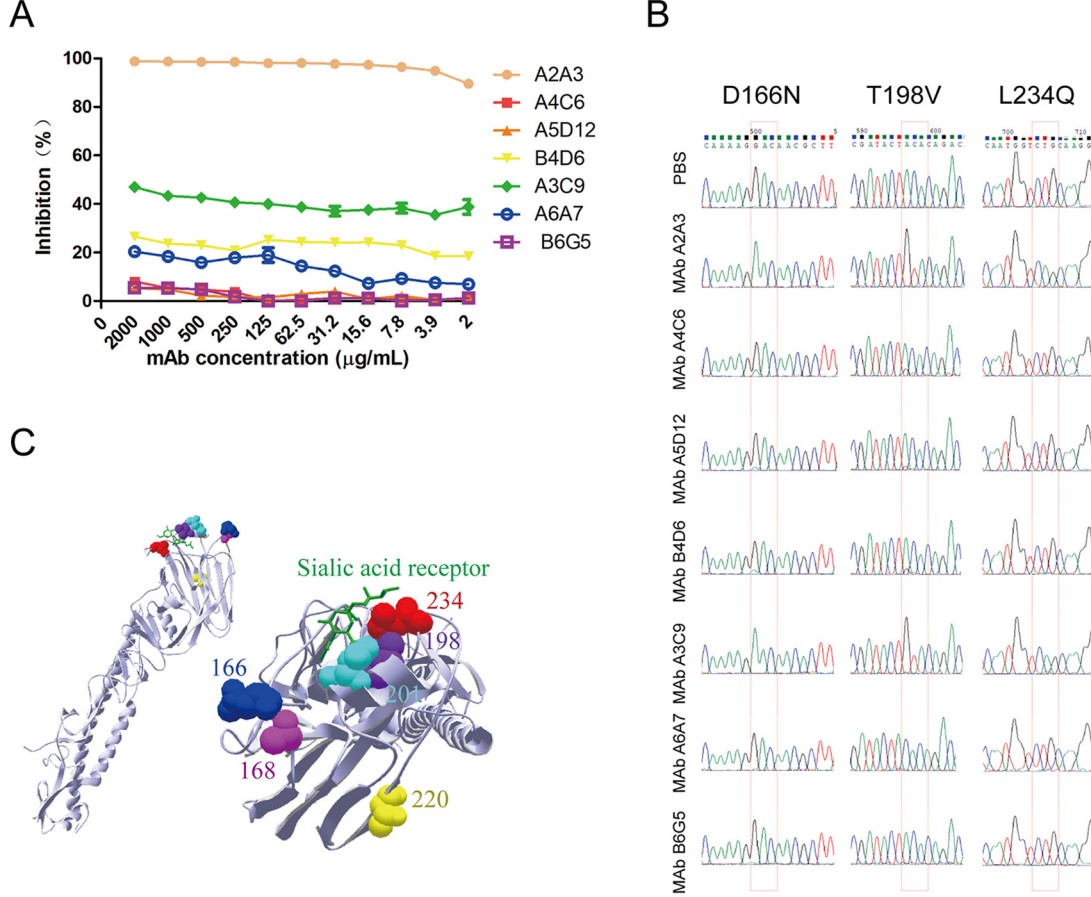

**FIG 1** NA antibody pressure resulted in mutations in the HA head. (A) NI abilities of N2-specific MAbs against WT XXM virus were tested in the Mu-NANA assay. Each concentration was tested in duplicates, and the data are represented as the mean values ± standard errors of the means (SEM). (B) HA mutations caused by incubation with N2-specific MAbs in chicken embryos were examined by HA sequencing and analyzed with Chromatogram software. Substitutions in nucleotide codons encoding the amino acids at positions 166, 198, and 234 are highlighted with rectangles. (C) A model of the locations of mutant residues on HA was generated by Swiss PdbViewer/DeepView with the HA monomer (PDB code 1JSH). The structure is shown in side view (left) and top view (right). The mutant residues and sialic acid receptor are marked with different colors on the HA head.

234, located in the receptor-binding pocket of HA, are crucial for binding with the sialic acid receptor, especially residues 198 and 234. The other residues, 166, 168, and 220, are located around the receptor-binding pocket but at a small distance from the direct binding sites.

**HA mutations caused by NA antibody pressure affected the thermostability and receptor-binding ability of HA.** The substitutions of N166 and T168 in mA3C9-1 and mA3C9-3 also introduced an *N*-linked glycosylation modification close to the receptor-binding pocket. However, the single mutation of T220I in mA2A3-4 removed an *N*-linked glycosylation modification on the HA head. The Western blotting result also proved that the HA protein of mA2A3-4 was slightly smaller in size than the WT HA protein (Fig. 2A), while mA3C9-1 and mA3C9-3 had slightly larger molecular weights because of the addition of a new *N*-linked glycosylation on the HA head. In the thermal stability test, we also found that mA2A3-4 was much more sensitive than the other viruses and lost hemagglutination ability after exposure at 56°C over 120 min (Fig. 2B). The other mutants still maintained high HA titers ($\geq 2^6$) after 180 min of incubation at 56°C.

The receptor-binding preference of each virus was measured by using $\alpha 2,3$-sialidase-treated chicken red blood cells (cRBCs) and a sialic acid competition assay (Table 2). The WT XXM virus had an HA titer of $2^{10}$ with $\alpha 2,3$-sialidase-treated cRBCs, as

**TABLE 1** HA mutations identified in escape mutants of N2-specific MAbs[a]

| Mutant | Mutation(s) in: | |
| | NA | HA |
| --- | --- | --- |
| mA2A3-1 | D369V | ND[b] |
| mA2A3-2 | E368K | D166N/T198V/N201D/L234Q |
| mA2A3-3 | R344K | ND |
| mA2A3-4 | R344I | T220I |
| mA4C6-1 | D369N | ND |
| mA4C6-2 | S400R | ND |
| mA4C6-3 | E368K | ND |
| mA5D12-1 | D369N | ND |
| mA5D12-2 | S400R | ND |
| mB4D6 | D369N | ND |
| mA3C9-1 | D125G/K296N | D166N/A168T/T198V/L234Q |
| mA3C9-2 | K296N | ND |
| mA3C9-3 | R253K | D166N/A168T/T198V/L234Q |
| mA6A7 | G248E | ND |

[a]The mutations in HA of the purified escape mutants were detected by HA sequencing.
[b]ND, not detected.

with untreated cRBCs. The WT virus still had an HA titer of $2^{10}$ after being preincubated with the disialoganglioside GD1a, which is an $\alpha$2,3-linked sialic acid receptor. All of the above-described results proved that the WT virus preferred to bind with $\alpha$2,6-linked sialic acid receptor but not $\alpha$2,3-linked sialic acid receptor. However, mA3C9-1 and mA3C9-3 did not agglutinate the $\alpha$2,3-sialidase-treated cRBCs, and both showed higher binding ability to $\alpha$2,3-linked sialic acid receptor in the sialic acid competition assay. mA2A3-2 and mA2A3-4 also showed higher binding ability to $\alpha$2,3-linked sialic acid receptors than the WT virus.

The WT XXM virus also presented a higher preference for Neu5Ac$\alpha$2-6Gal$\beta$1-4GlcNAc$\beta$–poly[$N$-(2-hydroxyethyl)acrylamide] ($\alpha$2,6SLN-PAA) in a solid-phase enzyme-linked immunosorbent assay (spELISA) (Fig. 2C). All mutants with HA mutations possessed lower affinity to $\alpha$2,6SLN-PAA but higher binding ability to Neu5Ac$\alpha$2-3Gal$\beta$1-4GlcNAc$\beta$-PAA ($\alpha$2,3SLN-PAA). The single mutation T220I in the HA of mA2A3-4 also switched the receptor-binding preferences from $\alpha$2,6-linked sialic acid receptor to $\alpha$2,3-linked sialic acid receptor.

All of the results show that inhibition of NA activity caused by NA antibody pressure would disrupt the HA-NA balance (Fig. 3). Moreover, the imbalance might have significant influence on HA and result in mutations in or around the RBS. In this study, the HA mutations switched the receptor-binding preference for $\alpha$2,6-linked sialic acid receptor to a preference for $\alpha$2,3-linked sialic acid receptor. These mutations may contribute to the survival of newborn viruses under NA antibody pressure.

**HA mutations promoted viral replication in MDCK cells and mouse lungs.** To assess the impact of the receptor-binding preference switch on the growth of the mutant viruses, we first examined the growth kinetics of each virus in Madin-Darby canine kidney (MDCK) cells. After infection at a low multiplicity of infection (MOI) of 0.01, the mA2A3-2 group reached the highest viral titer at 12 h, and all mutant viruses tested grew to higher peak titers than the WT virus at 48 h or 60 h (Fig. 4A).

The virulence of each mutant was further investigated in mice by intranasal inoculation of a median $10^5$ tissue culture infective dose (TCID$_{50}$) of virus per mouse. No evident weight loss was observed in any group except the mA2A3-2 group (Fig. 4B). However, all of the mutants were detected in the mouse lungs with viral titers that were higher but not significantly different from the viral titers in the mice challenged with WT virus at 3 days postinfection (Fig. 4C). By day 6, mice infected with mA2A3-2, which had combined mutations D166N/T198V/N201D/L234Q in HA, still had about 10-fold higher viral titers in the lungs than the group challenged with WT virus. Severe pathological lesions and inflammation in local areas of lungs were observed in all infected mice at day 6 postinfection (Fig. 4E). In addition, alveolar damage and inflammatory hyperplasia took place in the

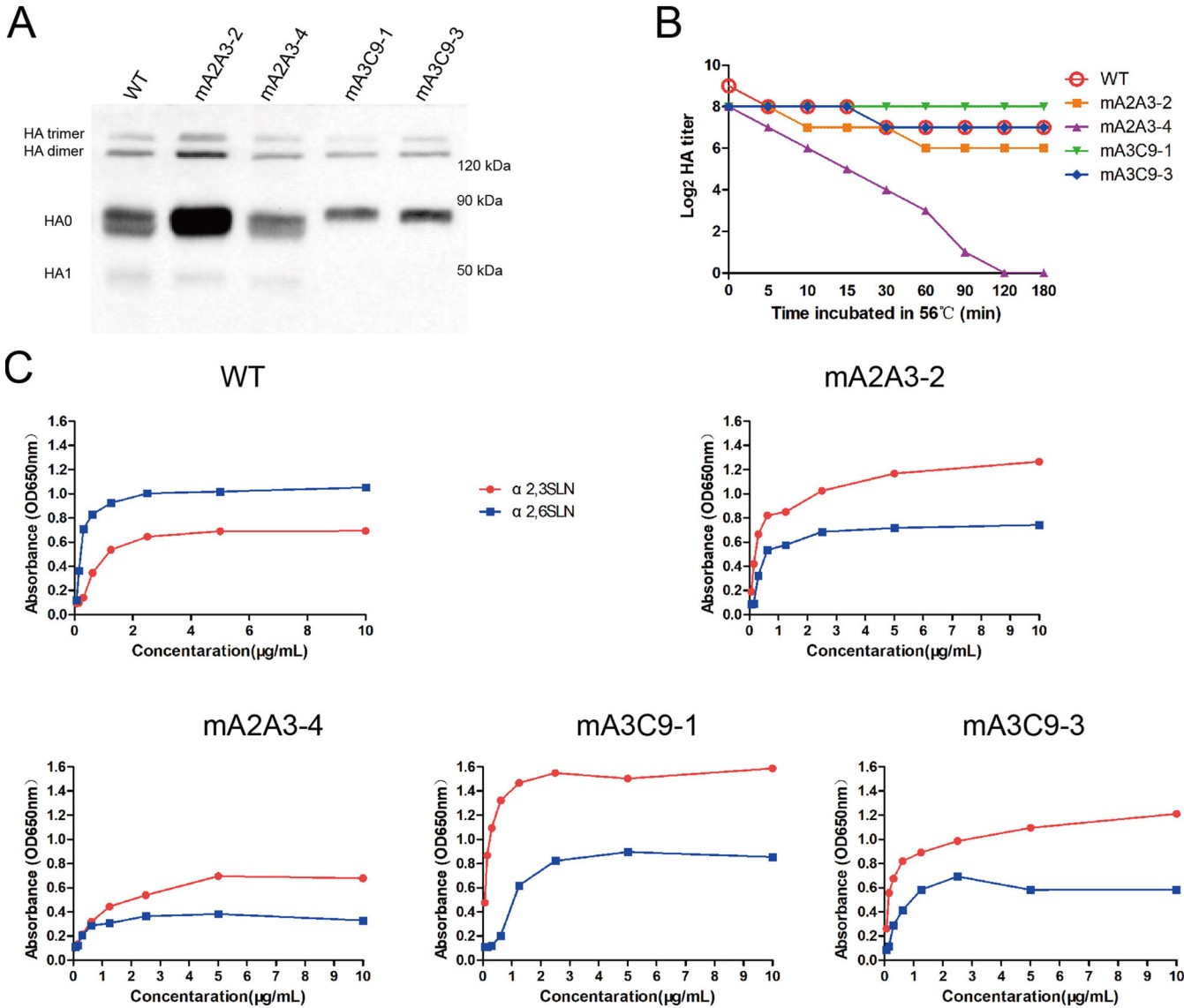

**FIG 2** HA mutations changed the thermal stability of HA and switched the receptor tropism. (A) Nonreducing allantoic fluid of each mutant virus was tested by Western blotting. HA trimer, HA dimer, HA0 (HA monomer), and HA1 are indicated on the left. (B) HA titers of the mutant viruses incubated at 56°C for different times were measured with 0.5% chicken red blood cells (cRBCs). (C) Two synthetic glycan receptors, $\alpha$2,3SLN-PAA-biotin and $\alpha$2,6SLN-PAA-biotin, were used to test the binding preference of the WT virus and mutant virus. Wells of 96-well plates were coated with serially diluted glycan receptors (10, 5, 2.5, 1.25, 0.62, 0.31, 0.15, and 0.07 $\mu$g/ml) and the plates further incubated with each virus. OD650nm, optical density at 650 nm.

lungs of all groups except the negative-control group challenged with phosphate-buffered saline (PBS). Extensive lung hemorrhage and serous effusion were observed in the lungs of the mA2A3-2 challenged mice, which may be strongly related to the continuous high viral replication level of mA2A3-2 in mouse lungs (Fig. 4C). A serum sample was also collected from each mouse at the end of the mouse experiment, and the hemagglutination inhibition (HI) titers against the WT virus and mutant viruses were measured in an HI assay (Fig. 4D). Interestingly, all sera showed better inhibition effects against the WT virus and mA2A3-4, but not against mA2A3-2 and mA3C9-3, which contained combined mutations. The HI results implied that HA mutations and glycosylation differences caused by NA antibodies resulted in significant antigenic changes in mutant HAs.

**NA antibody pressure is involved in HA antigenic evolution.** To further verify the effect of HA mutations in the antigenic change, previously prepared H9-specific MAbs and serum from chickens challenged with WT XXM virus were used to determine the HI titers of the escape mutants (21). Mutants mA2A3-2, mA3C9-1, and mA3C9-3 had

**TABLE 2** The effect of HA mutations on hemagglutination

| Virus | HA titer (log$_2$) against cRBCs that were[a]: | |
| | Treated[b] | Untreated |
| --- | --- | --- |
| **Untreated** | | |
| WT | 10 | 10 |
| mA2A3-2 | 7 | 8 |
| mA2A3-4 | 3 | 8 |
| mA3C9-1 | 0 | 11 |
| mA3C9-3 | 0 | 8 |
| | | |
| **Treated[c]** | | |
| WT | —[d] | 10 |
| mA2A3-2 | — | 0 |
| mA2A3-4 | — | 6 |
| mA3C9-1 | — | 5 |
| mA3C9-3 | — | 3 |

[a]cRBCs, chicken red blood cells.
[b]cRBCs were pretreated with α2,3-sialidase. The HA titer of the virus to the treated cRBCs indicates viral preference for α2,6-linked sialic acid receptor.
[c]Viruses were pretreated with GD1a. The decreased HA titer of the treated virus to untreated cRBCs indicates viral preference for α2,3-linked sialic acid receptor.
[d]—, not done.

lower HI titers for all H9-specific MAbs and chicken serum than the WT virus (Table 3), while mA2A3-4 virus showed a weaker decrease in HI titer for all H9-specific MAbs but had no effect on the chicken serum. All results indicate that combined mutations caused by NA antibodies caused antigenic changes in HA.

Our previous research proved that antibodies against NA were involved in antigenic drift in H9N2 IAVs in China from 1999 to 2019 (20). To identify whether NA antibody pressure is involved in HA antigenic evolution, the evolutionary correlations between

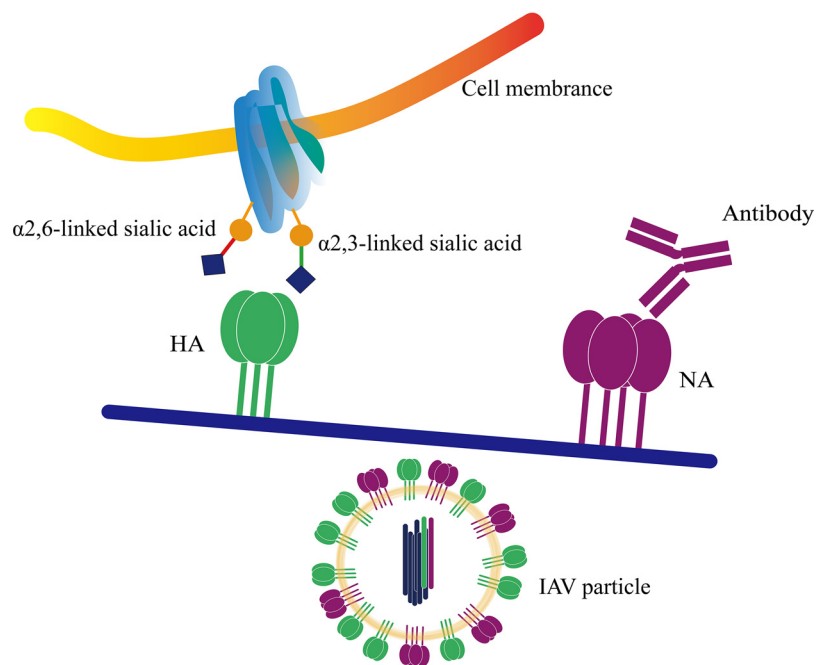

**FIG 3** NA antibody pressure breaks the HA-NA balance and affects receptor-binding preference. The sialic acid receptors are shown in the form of blue diamonds. The blue see-saw is used to show HA-NA function balance. The inhibition of NA by NA-specific antibodies results in dysfunction of NA, which further leads to the imbalance of HA-NA function. When the pressure transmits to the flexible HA, HA mutations take place in the HA head and change the receptor-binding preferences for survival under NA antibody pressure.

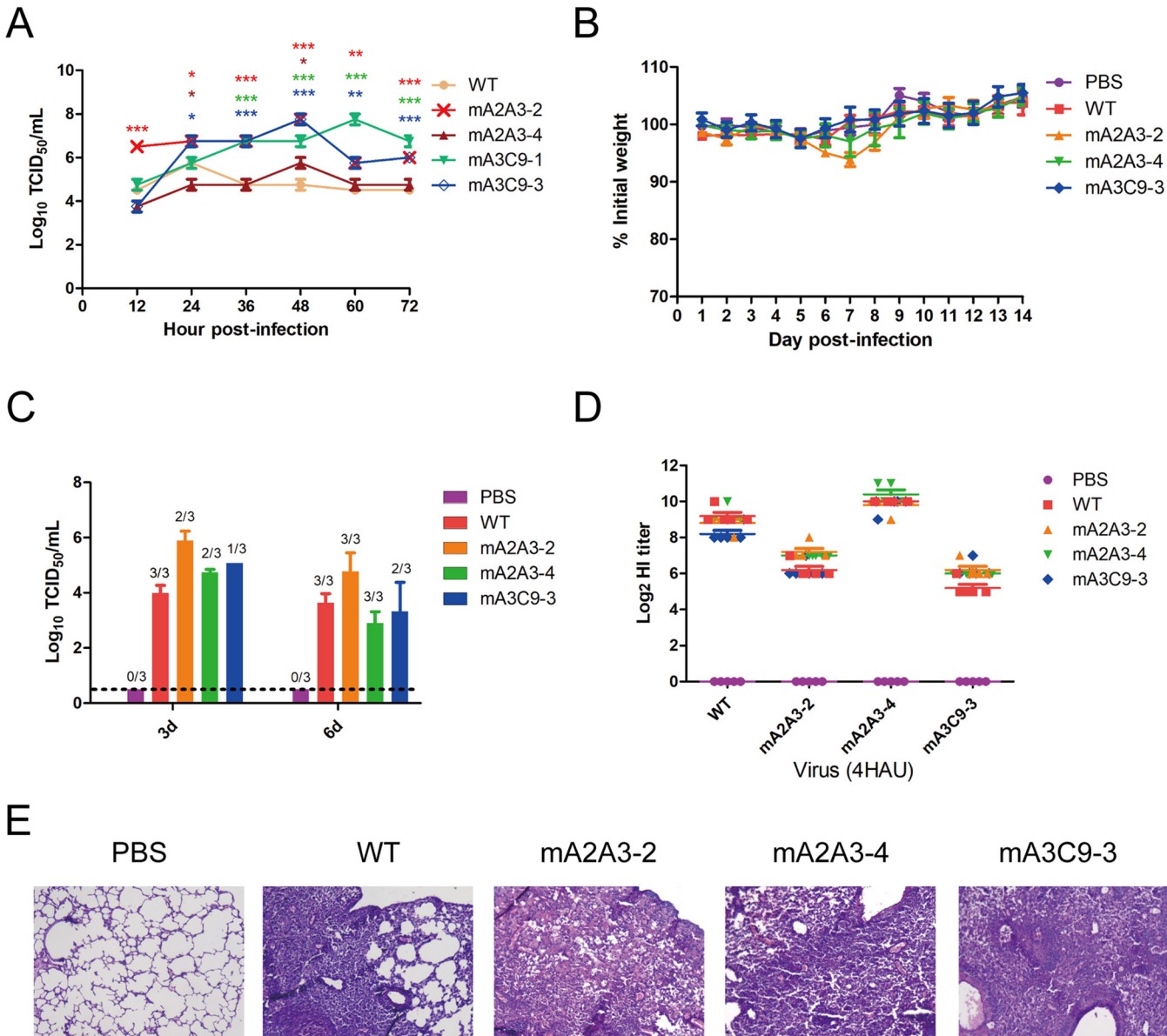

**FIG 4** HA mutations caused by NA antibody improve viral growth. (A) Viral growth kinetics of the WT XXM virus and the mutants in MDCK cells. All data are presented as the mean values ± SEM of three duplicates from two independent experiments. The data for the mutant viruses and the WT virus were compared by two-way ANOVA in GraphPad Prism 5 (*, $P < 0.05$; **, $P < 0.001$; ***, $P < 0.0001$). (B) Body weight changes of mice ($n = 5$ per group) infected with the WT XXM virus and the mutants. All data are represented as mean values ± SEM. (C) Viral loads in lungs from the mice on days 3 and 6 postinfection. Lungs of 3 mice were tested each time. The ratio of positive samples to total samples is shown above each column. (D) The HI titers of serum samples ($n = 5$ per group) cross-reacting with the WT XXM virus and mutants were measured in an HI assay. (E) Histological lesions in the lungs from the infected mice on day 6 postinfection. Representative images of hematoxylin and eosin (H&E)-stained lung tissues are shown in ×200 magnification.

NA and HA of H9N2 IAVs in China were analyzed online with Nextstrain (https://nextstrain.org/flu/avian/h9n2).

In the 1990s and 2000s, HA mutations V198A and Q234L and mutations of other antigenic sites in the HA head were mainly caused by host adaption and HA-specific antibodies induced by the inactivated H9N2 vaccines that were first used in poultry in China in the 1990s (Fig. 5). Similarly, mutations like R199K and K368E around the active center of NA can also be found in NA of the H9N2 virus in this period. After that, there were nearly no mutations at these sites in HA of the main branch of H9N2 IAVs from 2000 to 2010. However, an amino acid at position 368 in NA that is crucial for NA-specific neutralizing antibody was continuously changing, which was caused by strong pressure from

**TABLE 3** The effect of HA mutations on HI titers of H9-specific MAbs and chicken serum

| Virus | HI titer (log$_2$) of[a]: | | | | |
| | IAV-H9-2G4 | IAV-H9-6E6 | IAV-H9-5B4 | IAV-H9N2-chicken serum[b] | Control MAb |
|---|---|---|---|---|---|
| WT | 14 | 13 | 16 | 7 | ND[c] |
| mA2A3-2 | 11 | 11 | 11 | 5 | ND |
| mA2A3-4 | 13 | 12 | 15 | 7 | ND |
| mA3C9-1 | 11 | 11 | 12 | 5 | ND |
| mA3C9-3 | 11 | 11 | 12 | 5 | ND |

[a]The HI titers of antibodies were measured by incubation with 4 HAU virus for 15 min at 37°C and further reaction with 0.5% cRBCs.
[b]From chickens infected with WT XXM H9N2 virus.
[c]ND, not detected.

the presence of NA antibody in poultry during this decade. After 2010, mutations at positions 168, 198, 201, and 220 occurred in HA, which may have been caused by the NA antibody pressure, because the D368N and D369G mutations in NA have introduced an *N*-linked glycosylation modification around the NA active center, which is a significant marker of NA antibody pressure (20).

## DISCUSSION

Antigenic drift in HA and NA helps IAVs escape from humoral immunity in hosts and makes it necessary to continuously update vaccines. Mutations in a protein are always supposed to be induced by antibodies targeting it, whereas mutations caused by the HA-NA balance are rarely reported (22, 23). In this study, we have identified mutations in HA caused by NA antibody pressure. Furthermore, the higher antibody pressure would increase the levels of mutations at positions 166, 198, and 234 in HA of H9N2 viruses.

The RBS of HA consists of the 130 loop, 190 helix, and 220 loop (24). The receptor tropism of influenza virus also strongly affects the NA activity (10). Positions 198 and 234 (190 and 226 in H3 numbering) in the RBS are key sites for determining receptor-binding avidity. In particular, L234 is the marker for binding with human-type α2,3-linked sialic acid receptor, while Q234 is the marker for binding with avian-type α2,6-linked sialic

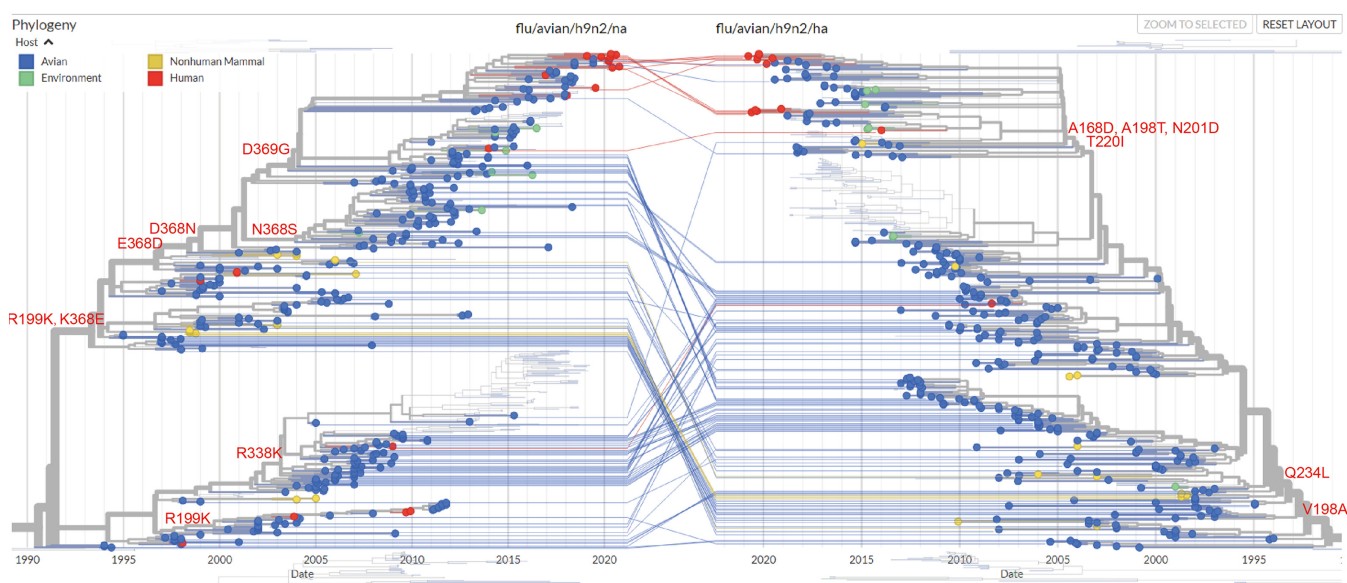

**FIG 5** The effect of NA antibody pressure on HA evolution. The major branch of the phylogenetic tree of full-length HA and NA genes of H9N2 IAVs isolated in China from 1994 to 2020 was generated with Nextstrain (https://nextstrain.org/flu/avian/h9n2). Viruses in the phylogenetic trees are colored according to the hosts. The NA mutations selected by antibody pressure are indicated as markers for NA (left). The HA mutations at positions 166, 168, 198, 201, 220, and 234 are indicated on the HA branch (right). HA and NA genes belonging to the same virus are connected by lines.

acid receptor (25, 26). Deglycosylation of position 166 (158 in H3 numbering) is important for the dual receptor-binding properties of H5 subtype avian influenza viruses, while glycosylation would decrease affinity to the human-type receptor (27–29). Consistent with this, the escape mutants with Q234 and N166 in this research also showed higher affinity to the avian-type receptor but weaker affinity to the human-type receptor. Glycosylation at site 166 (158 in H3 numbering) can enhance the pathogenicity of H5N1 and H5N6 viruses in mice (30, 31). In our experiment, glycosylation at site 166 in the HA of mutant H9N2 virus only resulted in significantly higher viral growth in MDCK cells and did not enhance viral pathogenicity in the mice.

Antibodies targeting the HA head, especially those that are close to the RBS in HA1, can prevent influenza virus attachment by steric hindrance and have HI ability (32, 33). However, mutations in the HA head and RBS variations would help virus evade these antibodies (21). Single or combined mutations at positions 166, 168, 198, 201, and 234 were involved in switching the antigenic phenotype and receptor specificity of seasonal H3N2 virus and H9N2 IAV (34–36). The HA mutations identified in this study also decreased the HI titers of serum samples from challenged mice, H9-specific MAbs, and chicken serum, which indicated that HA mutations caused by NA antibody pressure can also help virus evade neutralizing antibodies against HA. Moreover, the serum of challenged mice showed higher inhibition of viruses with fewer mutations and glycosylation sites, which revealed significant antigenic drift caused by NA-specific antibody pressure.

H9N2 vaccines manufactured with H9N2 viruses isolated in the 1990s are still widely used in poultry in China (37). The H9-specific antibodies induced by these vaccines in poultry were reported to provide a weak protective effect against current epidemic strains (38–40), while combined mutations A168D/A198T/N201D/T220I can be found in HA of H9N2 strains isolated after 2010, which is highly correlated with NA antibody pressure rather than HA antibody pressure. These mutations and the L234 in HA of the current field strains may be the main cause of the increase in human H9N2 infections in recent years.

NA antibodies are crucial for humoral immunity in natural infection and vaccination (41, 42). However, the inhibition of NA caused by existing antibody pressure would result in both NA and HA mutations. NA mutations can reduce the inhibition of NA-specific neutralizing antibodies, while the substitutions in HA would not only change the receptor tropism but also help virus escape from HA-specific protective antibodies. All in all, NA antibody pressure is a double-edged sword which should not be overlooked.

## MATERIALS AND METHODS

**Viruses, MAbs, and cells.** The WT XXM H9N2 influenza virus and escape mutants selected by N2-specific MAbs were propagated in 9-day-old and specific-pathogen-free (SPF) embryonated chicken eggs as previously reported (20). Allantoic fluid was collected on day 5 postinoculation and stored at −70°C for further study. Hybridomas that secret MAbs against the HA and NA of H9N2 influenza virus were prepared in previous research (20, 21). Ascitic fluid of each hybridoma was prepared in BALB/c mice. N2-specific MAbs (A2A3, A4C6, A5D12, B4D6, A3C9, A6A7, and B6G5) were all purified with a protein G column (GE, Shanghai, China) and stored at −70°C for further use. The MDCK cells were maintained in Dulbecco modified Eagle medium (DMEM; Gibco, Shanghai, China) supplemented with 10% fetal bovine serum (FBS) at 37°C in 5% $CO_2$.

**NI assay.** The NI activity of each MAb was measured by the Mu-NANA assay. The Mu-NANA assay was carried out according to a previous report (43). Briefly, mixtures of predetermined amounts of virus and serially diluted MAbs were incubated in a black 96-well plate for 30 min at 37°C. A volume of 50 $\mu$l of 0.2 mM Mu-NANA substrate (Sigma, Shanghai, China) was added to each well and incubated for 1 h at 37°C. The reaction was finally stopped with 0.2 M $Na_2CO_3$ and read by using a BioTek Synergy2 reader with an excitation range of 350 to 365 nm and an emission range of 440 to 460 nm.

**Escape mutant selection and HA sequencing.** The escape mutant selection was conducted as previously described (21). A volume of 50 $\mu$l of allantoic fluid of XXM virus was incubated with 0.5 ml MAb against NA or PBS at 37°C for 30 min and inoculated into five 9-day-old SPF embryonated eggs. Viral RNA was extracted from allantoic fluid with the FastPure cell/tissue total RNA isolation kit (Vazyme Biotech, Jiangsu, China). Amplification of the HA gene was carried out by reverse transcription-PCR (RT-PCR). PCR products were sequenced by BGI, Shanghai, China. Nucleotide and amino acid sequences were analyzed with Chromas software (Technelysium, Australia) and Lasergene software (DNASTAR, Inc., USA). The WT HA gene (accession number MZ144026) of the XXM H9N2 virus was used for alignment.

**Western blot analysis.** Allantoic fluid of each virus was used for nonreducing polyacrylamide gel electrophoresis as previously described (44). The allantoic fluids were treated with loading buffer without DL-dithiothreitol (DTT). Treated samples were used for sodium dodecyl sulfate-polyacrylamide gel electrophoresis (SDS-PAGE) and then transferred to nitrocellulose membranes (GE, MA, USA) for Western blot analysis. The H9-specifc MAb 2G4 was diluted 5,000-fold with PBS containing 0.5% Tween 20 (PBST) and used as the primary antibody for HA protein determination. After washing six times with PBST, peroxidase-conjugated goat anti-mouse IgG(H+L) antibody (Jackson ImmunoResearch, PA, USA) was diluted 10,000-fold and used as the secondary antibody. After washing another six times with PBST, the membrane was finally immersed in Immun-Star horseradish peroxidase (HRP) substrate (Bio-Rad, CA, USA), and the chemiluminescent signals were observed with the FluorChemE imaging system (Protein Simple, CA, USA).

**Thermal stability test.** Allantoic fluid of each virus was incubated in a 56°C water bath for different times (0, 5, 10, 15, 30, 60, 90, 120, and 180 min). The HA titers of heat-treated viruses were then determined with 0.5% cRBCs.

**Receptor-binding assay.** The receptor-binding specificity of the virus to $\alpha2,6$-linked sialic acid receptor was determined by comparing the hemagglutination titers between the native and $\alpha2,3$-linked-sialidase-treated cRBCs as described previously (27). PR8 virus was used as the control. Briefly, 3 ml of a 10% suspension of cRBCs was incubated with 0.5 U $\alpha2,3$-sialidase (TaKaRa, Beijing, China) for 1.5 h at 37°C. After washing 3 times with PBS, the treated cRBCs were adjusted to a final concentration of 0.5% with PBS. The hemagglutination titer of each virus was determined with the treated cRBCs. The receptor-binding specificity of the virus to $\alpha2,3$-linked sialic acid receptor was measured in a sialic acid competition assay. The disialoganglioside GD1a (GlycoSci, Shanghai, China), which contains $\alpha2,3$-linked sialic acid receptor, was used to competitively bind virus. Briefly, allantoic fluid of each virus was first treated with 25 $\mu$M/ml zanamivir for 1 h at 37°C. Then, the virus was further incubated with 12.5 $\mu$g/ml GD1a overnight at 4°C. After incubation, the GD1a-treated virus and untreated virus were finally tested with normal 0.5% cRBCs.

**spELISA.** The receptor-binding preference of each mutant virus was further measured with Neu5Ac$\alpha2$-3Gal$\beta1$-4GlcNAc$\beta$–poly[N-(2-hydroxyethyl)acrylamide]-biotin ($\alpha2,3$SLN-PAA-biotin) and Neu5Ac$\alpha2$-6Gal$\beta1$-4GlcNAc$\beta$–PAA-biotin ($\alpha2,6$SLN-PAA-biotin) (GlycoTech, MD, USA) based on a previous report (27). Serially diluted glycan receptors were added into Pierce streptavidin-coated high-binding-capacity 96-well plates (Thermo Fisher, IL, USA) and incubated overnight at 4°C. The coated wells were further blocked with PBST containing 5% skim milk powder and incubated with allantoic fluid of each virus (HA titer $>2^6$) pretreated with 25 $\mu$M/ml zanamivir. Then, mouse serum against WT XXM virus was used as the primary antibody and peroxidase-conjugated goat anti-mouse IgG(H+L) (Jackson ImmunoResearch, PA, USA) was used as the secondary antibody. The plates were washed with PBST another six times, followed by the addition of tetramethyl benzidine (TMB) substrate. The reaction was stopped with 1% SDS, and the absorbance at 650 nm was read.

**Viral growth kinetics.** MDCK cells growing in 6-well plates were infected with each mutant virus at a MOI of 0.001. The supernatants of the infected cells were collected at 12, 24, 36, 48, 60, and 72 h postinfection, the viruses were titrated in MDCK cells, and $TCID_{50}$ values were calculated by Reed-Muench assay (45).

**Mouse experiment.** Three mutants and the WT XXM H9N2 virus were used for the mouse study. Six-week-old BALB/c mice purchased from Experimental Animal Center of Yangzhou University (Yangzhou, China) were divided into 5 different groups (11 mice per group). The mice were anesthetized with 0.2 ml 1.25% avertin by intraperitoneal injection and challenged with $10^5$ $TCID_{50}$ of H9N2 virus in 50 $\mu$l PBS by intranasal inoculation (46). The control group was administered PBS. Three mice in each group were euthanized on days 3 and 6 postinfection, and the lungs were collected for virus titration. The lung tissues at day 6 postinfection were collected and fixed with 4% formaldehyde for histopathological examination. The remaining 5 mice in each group were monitored for clinical signs and weight loss for 14 days, and sera were collected at the end of the experiment.

**HI assay.** The HI assay was performed with 4 hemagglutination units (HAU) of virus following a previous report (21). Ascitic fluid of H9-specific MAbs, chicken serum, and mouse serum were serially diluted and mixed with predetermined virus. Chicken serum against XXM virus was prepared in 2-week-old SPF chickens challenged with $10^6$ $TCID_{50}$ virus by nasal drip and collected at 14 days postinfection. MAb 1D10 against the fusion (F) protein of Newcastle disease virus (NDV) was used as the negative control. After incubation for 15 min at 37°C, the mixtures were tested with 0.5% cRBCs.

**Statistical analysis.** All data were analyzed with GraphPad Prism version 5 (www.graphpad.com) and are presented as the mean values $\pm$ standard errors of the means (SEM). The viral growth titers were compared by two-way analysis of variance (ANOVA) (*, $P < 0.05$; **, $P < 0.001$; ***, $P < 0.0001$).

**Ethical approval.** For mouse experiments, 6-week-old female BALB/c mice were purchased from the Experimental Animal Center of Yangzhou University (Yangzhou, China). All animal experiments were done in accordance with the institutional animal care guidelines, and the protocol (number 06R015) was approved by the Animal Care Committee of Yangzhou University.

## ACKNOWLEDGMENTS

This work was supported by the National Key Research and Development program of China (grant number 2017YFD0501100) and the Priority Academic Program Development of Jiangsu Higher Education Institutions.

F.W., Z.W., and A.Q. designed the work. F.W., Y.W., J.W., and H.F. conducted experiments. F.W., Z.W., K.Q., H.S., A.Q., and J.Y. analyzed the data. H.F. and W.G.

designed, conducted, and analyzed pathological analysis of mouse lungs. F.W. and Z.W. wrote the manuscript. All authors reviewed and edited the paper.

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
