## [Reviewer comments · Microbiology Spectrum]

Microbiology Spectrum

Identification of haemagglutinin mutations caused by neuraminidase antibody pressure

Fei Wang, Zhimin Wan, Yajuan Wang, Jinsen Wu, Hui Fu, Wei Gao, Hongxia Shao, Kun QIAN, Jianqiang Ye, and Aijian Qin

Corresponding Author(s): Aijian Qin, Yangzhou University

Review Timeline:

Submission Date:	September 1, 2021
Editorial Decision:	October 26, 2021
Revision Received:	November 25, 2021
Accepted:	November 30, 2021

Editor: Heba Mostafa

Reviewer(s): Disclosure of reviewer identity is with reference to reviewer comments included in decision letter(s). The following individuals involved in review of your submission have agreed to reveal their identity: Hongjun Chen (Reviewer #2)

Transaction Report:

DOI: <https://doi.org/10.1128/spectrum.01439-21>

October 26, 2021

Dr. Aijian Qin
Yangzhou University
College of Veterinary Medicine
12 East wenhui road
Yangzhou 225009
China

Re: Spectrum01439-21 (Identification of haemagglutinin mutations caused by neuraminidase antibody pressure)

Dear Dr. Aijian Qin:

Thank you for submitting your manuscript to Microbiology Spectrum. When submitting the revised version of your paper, please provide (1) point-by-point responses to the issues raised by the reviewers as file type "Response to Reviewers," not in your cover letter, and (2) a PDF file that indicates the changes from the original submission (by highlighting or underlining the changes) as file type "Marked Up Manuscript - For Review Only". Please use this link to submit your revised manuscript - we strongly recommend that you submit your paper within the next 60 days or reach out to me. Detailed information on submitting your revised paper are below.

Link Not Available

Sincerely,

Heba Mostafa

Journals Department
Reviewer comments:

Reviewer #1 (Comments for the Author):

HA/NA balance is an important question in the influenza field as it controls viral production and interspecies transmission and adaptation. This manuscript by Wang et al. directly follows a previous study of the group describing NA antigenic changes of H9N2 virus and characterizing anti-N2 monoclonal antibodies. This new study aims at characterizing mutant viruses that escaped anti-N2 antibody pressure. The authors showed that NA antibody pressure caused the appearance of mutations in the HA in the RBS domain, hence affecting thermostability, receptor binding specificity and viral growth in vitro. The authors designed rigorous experiments and gave adequate interpretations and conclusions. Enough background is provided along with discussions of the data. The manuscript is well written and easy to follow. A few minor points (stated below) would improve the manuscript.

Minor comments:

- The influenza subtype (H9N2) and the species should be mentioned in the abstract.
- L27. What do "these variants" refer to? Please explain in the text.
- Figure could be improved by adding the specific amino acid substitutions directly on the chromatograms and on the structure.

- Crucial information is missing in the tables. Please describe the experiments done directly in the table legends.
- The full name of the XXM strain should be explicitly stated when it is first introduced.
- Fig2B and Fig4D: Some lines are overlaid by others and difficult to see. Can this be improved?
- Fig2C should use the same scale for all y-axis graphs.
- Fig3 is not clear. I believe it is dispensable or should be substantially improved.
- Fig4. The authors should describe in the legend how many experiments were performed with the number of replicates (n).
- Fig4A. What is the post hoc test used for multiple comparisons after 2-way ANOVA? It should be stated in the legend and in the methods section.
- Fig4C. Please specify in the figure legend what the numbers on top of the bars refer to.
- L185. Please provide a reference.
- L133. The authors should conclude the results carefully. This statement needs to be toned down since the data is not significant in mice.
- L275. Please check that the correct reference is inserted (17?).
- L349. Is it 3dpi (in fig) or 4dpi (in method)?
- L344 and L356. Please double check the correct information is written.

Reviewer #2 (Comments for the Author):

This manuscript introduced an interesting story about HA mutations in H9N2 AIV caused by NA antibody pressure. HA mutations were found in escape mutants selected by N2-specific MAbs. These HA mutations changed the biological characteristics of the HA protein including the glycosylation, thermostability, receptor preferences and antigenic structure. However, some questions should be addressed before publication.

Major questions:

1. The mutations in HA and NA were monitored only by RT-PCR assay, while it is unknown if the NA mutations or HA mutations take place firstly under antibody pressure.
2. The HA mutations at the HA RBS like positions 198 and 234 are well understood, but how the mutations at positions 166 and 220 affect the receptor preference.
3. In mouse experiment, severe lung lesion post challenge can be detected, but no significant weight loss. What leads to these different results?

Minor questions :

1. Why the receptor binding ability measured in HA assay and spELISA was different?
2. The reference 37 does not confirm to the description at line 239.
3. Please check the abbreviation of the wild-type at line 133. The WT has been used at line 125. The other abbreviations should also be checked carefully.
4. Please check the format of reference 28. The authors' names are not correct.
5. The abbreviation mAb at line 286 does not comply with the ASM regulations. Please check all other abbreviations carefully.

Reviewer #3 (Comments for the Author):

The authors characterized several escape mutants of an H9N2 virus selected by NA-specific mAbs. They found some of these escape mutants carry mutations in HA, which may have impacted the HA receptor preference, thermal stability, and antigenicity in in vitro assays, as well as viral load in the mouse lungs. These findings are consistent with some previous reports with other subtypes of influenza viruses and are of some significance for elucidating the evolution of H9N2 viruses under the antibody selective pressure.

Major concerns

1. The authors have recently published another paper describing the escape mutants selected by a panel of mAbs (Wang et al., EMI, 2021), including those that were used in the current study. It should be clarified whether the mutations in HA and NA reported in the current study are identical to, or different from, those identified in the published research.
2. The authors should thoroughly edit the manuscript, especially to provide necessary details, so that the readers can follow and understand what the authors have done and what the authors have found. For instance in the Abstract: Which subtype of virus has been used in your study? What do "these variants" in line 27 (page 2) mean? How were these variants obtained/generated? Why the HA thermal stability and evolution analysis data were not described in the Abstract? Another example is by reading the first paragraph in the Results section, I do not know what these antibodies are and what virus you have been working on.
3. The authors have made multiple statements without any evidence or supporting data, which need to be corrected or modified to improve the manuscript. Here are some examples. 1) On pages 9-10 lines 196-199, the authors stated that "In the 1990s and 2000s, the HA mutations as V198A, Q234L and the other antigenic sites in HA head were mainly caused by host adaptation and HA-specific antibodies induced by the inactivated H9N2 vaccines which were firstly used in China poultry in 1990s (Figure 5)." How do you know these mutations were mainly caused by host adaptation and vaccination? 2) Page 10 lines 208-210, the authors stated that "It should also be noted that the increase of human H9N2 infections in China in recent years may correlate with these HA mutations caused by NA antibody pressure." In fact the human infection with H9N2 virus is still rare and there is not obvious increase in the human cases, and more importantly, with the switch in the receptor preference of the mutants from

alpha 2, 6 to alpha 2, 3, I would argue that these viruses would have a decreased ability to infect humans. 3) Page 11 lines 239-240: "The HA mutations identified in this study also decrease the binding ability of serum samples of challenged mice..." Did you measure the binding ability?

Minor comments

1. Page 5 lines 103-105: "The other MAbs A4C6, A5D12, B4D6 and A6A7 showed weaker NI activity in Mu-NANA assay, while still resulted in slight codon changes at same positions." What do you mean by "slight codon changes"? Did these also occur in HA? If yes, please specify these codon changes and the amino acid substitutions.
2. Pages 5-6 lines 110-111: "Moreover, HA mutations were also identified in 4 of 10 escape mutants purified by plaque assay before (Table 1)..." Were data displayed in Table 1 from the current study or a study published previously?
3. Page 7 lines 149-150: "All results show that inhibition of NA activity caused by NA antibody pressure would break the HA-NA balance (Figure 3)." There are no data to support that the HA-NA was off balance. How do you define the HA-NA balance? Do these mutants have enhanced/decreased NA activity or hemagglutination ability?
4. Page 8 lines 114-115: "however, all the mutants were detected at higher titers although not significant in mouse lungs at 3 days post-infection (Figure 4C)." What do you mean by "not significant"? Compared with what?
5. Pages 9-10 lines 182-210. While the authors were emphasizing the correlation between the HA mutations with NA antibodies, please do note that mutations at H9 positions 168 and 201 can also be selected by HA-specific antibodies as the authors' group has reported previously (Wan et al., JVI, 2014).
6. Page 14 lines 296-301. Please provide details about the western blot analysis, including the amount of each virus loaded, the treatment and conditions (with or without reductant, the buffer used, etc.) for the SDS-PAGE, the transfer of the viral proteins to the film, and the antibody used for probing the target and the concentration of the antibody, etc.
7. Page 16 lines 350-351: "...on the days 4 and 6 post-infection, and the lungs were collected for virus titration. The lung tissues were collected at day 6 post-infection and fixed with 4% formaldehyde for histopathological examination." Please specify how you have managed to divide each of the day 6 lung samples for viral titration and histopathological examination.
8. Please change "a" and "b" in the receptor analogs to α (alpha) and β (beta), wherever appropriate.
9. The authors are wished to be cautious in stating throughout the text that the HA mutations were caused by NA antibodies or NA antibody pressure. This may give the readers the impression that NA antibodies directly interacted with HA in your assay. I believe that the HA mutations were introduced to adapt to the NA mutations.
10. Tables 1-3, please provide necessary information including the viruses used in your study (for instance, what does "WT" mean?) and the source of antibodies/sera for your analyses.
11. Figure 3. Please explain more in the figure legend so that the readers can understand what message you want to convey through the schematic representation.

Staff Comments:

Preparing Revision Guidelines

Please return the manuscript within 60 days; if you cannot complete the modification within this time period, please contact me. If you do not wish to modify the manuscript and prefer to submit it to another journal, please notify me of your decision immediately so that the manuscript may be formally withdrawn from consideration by Microbiology Spectrum.

Corresponding authors may join or renew ASM membership to obtain discounts on publication fees. Need to upgrade your

membership level? Please contact Customer Service at Service@asmusa.org.

Reviewer comments:

Reviewer #1 (Comments for the Author):

HA/NA balance is an important question in the influenza field as it controls viral production and interspecies transmission and adaptation. This manuscript by Wang et al. directly follows a previous study of the group describing NA antigenic changes of H9N2 virus and characterizing anti-N2 monoclonal antibodies. This new study aims at characterizing mutant viruses that escaped anti-N2 antibody pressure. The authors showed that NA antibody pressure caused the appearance of mutations in the HA in the RBS domain, hence affecting thermostability, receptor binding specificity and viral growth in vitro.

The authors designed rigorous experiments and gave adequate interpretations and conclusions. Enough background is provided along with discussions of the data. The manuscript is well written and easy to follow. A few minor points (stated below) would improve the manuscript.

Response : Thanks for your kind summary.

Minor comments:

- The influenza subtype (H9N2) and the species should be mentioned in the abstract.

Response: Thank you for the advice. The description in the abstract has been modified at line 26-29 as following:

“In this study, we examined the presence of amino acid mutations in the HA of NA-specific antibodies selected H9N2 mutants and compared the HA properties to that of the wild-type (WT) A/Chicken/Jiangsu/XXM/1999 (XXM) virus.”

- L27. What do "these variants" refer to? Please explain in the text.

Response: Thank you for the question. The "these variants" has been changed into “NA-specific antibodies selected H9N2 mutants”, which is easier to understand.

- Figure could be improved by adding the specific amino acid substitutions directly on the chromatograms and on the structure.

Response: Thank you for the advice. The specific amino acid substitutions have been added on the chromatograms in Figure 1. However, the specific amino acid substitutions are not very suitable for marking on the structure. Each residue only represents its location but not specific amino acid.

- Crucial information is missing in the tables. Please describe the experiments done directly in the table legends.

Response: Thank you for the advice. The following experimental details were added in the legends.

“a: The mutations in HA of the purified escape mutants were detected by HA sequencing.”

“a: The cRBCs pre-treated with α 2,3-sialidase. The HA titer of the virus to the treated cRBCs indicated viral preference for α 2,6-linked sialic acid receptor.

b: Viruses pre-treated with GD1a. The decreased HA titer of the treated virus to untreated cRBCs

indicated viral preference for α 2,3-linked sialic acid receptor..”

“a: The HI titers of antibodies were measured by incubation with 4 HAU virus for 15 min at 37 °C and further reaction with 0.5% cRBCs.”

- The full name of the XXM strain should be explicitly stated when it is first introduced.

Response: Thank you for the advice. The full name has been explicitly stated at line 89.

- Fig2B and Fig4D: Some lines are overlaid by others and difficult to see. Can this be improved?

Response: Thanks. We have changed the symbols on the line in the Fig2B and Fig4D. To make it clear, the graphs in Fig4D were also showed bigger than before.

- Fig2C should use the same scale for all y-axis graphs.

Response: Thank you for the advice. The y-axis in Fig2C has been changed into same one ranging from 0 to 1.6.

- Fig3 is not clear. I believe it is dispensable or should be substantially improved.

Response: Thanks for your advice. The graph has been modified. More information has been provided in legend as following:

“The sialic acid receptors are shown in the form of blue diamond. The blue see-saws are used to show HA-NA function balance. The inhibition of NA by NA-specific antibodies results in dysfunction of NA which further lead to the imbalance of HA-NA function. When the pressure transmits to the flexible HA, HA mutations take place in HA head and change receptor binding preferences for survival under NA antibody pressure.”

.

- Fig4. The authors should describe in the legend how many experiments were performed with the number of replicates (n).

Response: Thank you for the advice. The description has been modified as following:

“Figure 4. HA mutations caused by NA antibody improve viral growth. (A) Viral growth kinetics of the WT XXM virus and the mutants in MDCK cells. All data were represented as means \pm SEM of three duplicates from two independent experiments. The data of the mutant viruses were compared with WT virus by two-way ANOVA in GraphPad prism 5 (*, P<0.05; **, P<0.001; ***, P<0.0001). (B) Body weight changes of mice (n=5 per group) infected with the WT XXM virus and the mutants. All data were represented as means \pm SEM. (C) Viral load in lungs from the mice on days 3 and 6 post-infection. Three mice lungs were tested each time. The ratio of positive samples to total samples was shown above the column. (D) The HI titers of each serum samples (n=5 per group) cross-reacting with the WT XXM virus and mutants were measured in HI assay. (E) Histological lesions in the lungs from the infected mice on day 6 post-infection. Representative images of H.E- stained lung tissues were shown in 200-fold magnification.”

- Fig4A. What is the post hoc test used for multiple comparisons after 2-way ANOVA? It should be stated in the legend and in the methods section.

Response: Thanks for your advice. We have modified as following:

“(A) Viral growth kinetics of the WT XXM virus and the mutants in MDCK cells. All data were

represented as means \pm SEM of three duplicates from two independent experiments. The data of the mutant viruses were compared with WT virus by two-way ANOVA in GraphPad prism 5 (*, $P < 0.05$; **, $P < 0.001$; ***, $P < 0.0001$).”

- Fig4C. Please specify in the figure legend what the numbers on top of the bars refer to.

Response: Thanks for your advice. We have modified as following:

“(C) Viral load in lungs from the mice on days 3 and 6 post-infection. Three mice lungs were tested each time. The ratio of positive samples to total samples was shown above the column.”

- L185. Please provide a reference.

Response: Thank you for the advice. We are very sorry that the chicken serum is prepared in another study which is still not published. We provide following detailed information in Materials and methods part at line 357.

“Chicken serum against XXM virus was prepared in two-week-old SPF chickens challenged with 10^6 TCID50 virus by nasal drip and collected at 14 days post infection.”

- L133. The authors should conclude the results carefully. This statement needs to be toned down since the data is not significant in mice.

Response: Thank you for the advice. We have changed the conclusion as following:

“In our experiment, glycosylation at site 166 in HA of mutant H9N2 virus only resulted in significantly higher viral growth in MDCK cells but not enhanced viral pathogenicity to the mice.”

- L275. Please check that the correct reference is inserted (17?).

Response : Thanks for your suggestion. We are sorry for the mistake here. It should be the reference 21 here. You can find it in the revised manuscript.

- L349. Is it 3dpi (in fig) or 4dpi (in method)?

Response : Thanks for your suggestion. We are sorry for the mistake here. It is 3dpi here. We have modified it in the manuscript.

- L344 and L356. Please double check the correct information is written.

Response : Thanks for your suggestion. We are sorry for the mistake here. It is 6-week-old BALB/c mice at line 356. The 8-week-old BALB/c mice were used for preparing ascitic fluid. We have modified it in the manuscript.

Reviewer #2 (Comments for the Author):

This manuscript introduced an interesting story about HA mutations in H9N2 AIV caused by NA antibody pressure. HA mutations were found in escape mutants selected by N2-specific MAbs.

These HA mutations changed the biological characteristics of the HA protein including the glycosylation, thermostability, receptor preferences and antigenic structure. However, some questions should be addressed before publication.

Response: Thanks for your comments.

Major questions:

1. The mutations in HA and NA were monitored only by RT-PCR assay, while it is unknown if the NA mutations or HA mutations take place firstly under antibody pressure.

Response : We have detected HA mutations without NA mutations in viruses incubated with MAb A3C9 by RT-PCR assay. While the HA and NA mutations in viruses incubated with other N2-specific MAb took place at the same time. The N2-specific antibody pressure led to the dysfunction of NA, which further caused HA mutations. We believe HA mutations take place first under high NA antibody pressure.

2. The HA mutations at the HA RBS like positions 198 and 234 are well understood, but how the mutations at positions 166 and 220 affect the receptor preference.

Response : Thanks for your question. As shown in Fig1C, residues 198 and 234 locate in the receptor binding pocket of HA, while positions 166 and 220 are little far away from the binding sites. The influence of the N-linked glycosylation at position 166 (158 in H3 numbering) on receptor preference has also been reported in H3 and H5 subtype viruses (DOI:10.1186/s13567-020-00879-6, DOI:10.3389/fmicb.2020.01318, DOI : 10.1128/JVI.01512-16). The N-linked glycosylation at position 220 is a novel site which may affect the HA binding preferences in similar way, because 220 loop is also crucial for receptor binding (DOI:10.1128/JVI.00218-17).

3. In mouse experiment, severe lung lesion post challenge can be detected, but no significant weight loss. What leads to these different results?

Response : Thanks for your question. As mentioned in manuscript, severe pathological lesions and inflammations were observed in local area of lungs not in the whole lungs. H9N2 AIV is a kind of low pathogenic virus and not lethal to animals in most cases.

Minor questions :

1. Why the receptor binding ability measured in HA assay and spELISA was different?

Response : Thanks for your question. The slightly differences between results of HA assay and spELISA are mainly caused by the different structure of the receptors and sensitivity of the assays. As reported in many studies, the viruses showed diverse affinity to the same type receptors with different structures (DOI: 10.1038/s41564-021-00976-y, DOI: 10.1038/nbt0909-797). The receptor binding ability measured in HA assay and spELISA was slightly different, but both

proved the switch of the receptor preferences.

2. The reference 37 does not confirm to the description at line 239.

Response : Thanks for your question. We have modified the description as following.

“Single or combined mutations at position 166, 168, 198, 201 and 234 were involved in switching the antigenic phenotype and receptor specificity of seasonal H3N2 virus and H9N2 IAV (35-37).”

3. Please check the abbreviation of the wild-type at line 133. The WT has been used at line 125. The other abbreviations should also be checked carefully.

Response : Thanks for your question. We have checked all abbreviations carefully. The sentence at line 133 has been revised as following:

“The WT virus still had an HA titer of 2^{10} after pre-incubated with a disialoganglioside GD1a which is an $\alpha 2,3$ -linked sialic acid receptor.”

4. Please check the format of reference 28. The authors' names are not correct.

Response: Thanks for your question. We have modified it in the manuscript. “Gao Y, Zhang Y, Shinya K, Deng G, Jiang Y, Li Z, Guan Y, Tian G, Li Y, Shi J, Liu L, Zeng X, bu Z, Xia X, Kawaoka Y, Chen H. 2009. Identification of Amino Acids in HA and PB2 Critical for the Transmission of H5N1 Avian Influenza Viruses in a Mammalian Host. PLoS pathogens 5:e1000709.”

5. The abbreviation mAb at line 286 does not comply with the ASM regulations. Please check all other abbreviations carefully.

Response: Thanks for your question. We have checked all abbreviations carefully. The sentence at line 286 has been revised as following:

“A volume of 50 μ L allantoic fluid of XXM virus was incubated with 0.5mL MAb against NA or PBS at 37 °C for 30 min and inoculated into five 9-day-old specific-pathogen-free (SPF) embryonated eggs”

Reviewer #3 (Comments for the Author):

The authors characterized several escape mutants of an H9N2 virus selected by NA-specific mAbs. They found some of these escape mutants carry mutations in HA, which may have impacted the HA receptor preference, thermal stability, and antigenicity in in vitro assays, as well as viral load in the mouse lungs. These findings are consistent with some previous reports with other subtypes of influenza viruses and are of some significance for elucidating the evolution of H9N2 viruses under the antibody selective pressure.

Response: Thanks for your comments.

Major concerns

1. The authors have recently published another paper describing the escape mutants selected by a

panel of mAbs (Wang et al., EMI, 2021), including those that were used in the current study. It should be clarified whether the mutations in HA and NA reported in the current study are identical to, or different from, those identified in the published research.

Response: Thanks for your question. In our previous work, we only analyzed the NA mutations involved in NA antigenic drift. Here, we mainly focused on the HA mutations selected by NA antibody pressure. The escape mutants in the Table 1 were all selected and purified by plaque assay in our previously published work. We further identified HA mutations in 4 of these escape mutants in this work. These escape mutants were further used in WB, thermal stability, receptor preferences and HI assays.

2. The authors should thoroughly edit the manuscript, especially to provide necessary details, so that the readers can follow and understand what the authors have done and what the authors have found. For instance in the Abstract: Which subtype of virus has been used in your study? What do "these variants" in line 27 (page 2) mean? How were these variants obtained/generated? Why the HA thermal stability and evolution analysis data were not described in the Abstract? Another example is by reading the first paragraph in the Results section, I do not know what these antibodies are and what virus you have been working on.

Response: Thanks for your question. We are sorry for the missing of the detailed information in the manuscript. We modified the abstract as following:

“The function balance of haemagglutinin (HA) and neuraminidase (NA) plays an important role in influenza virus genesis. However, whether and how N2-specific antibodies may impact the HA attributes remain to be investigated. In this study, we examined the presence of amino acid mutations in the HA of N2-specific monoclonal antibodies (MAbs) selected mutants and compared the HA properties to that of the wild-type (WT) A/Chicken/Jiangsu/XXM/1999 (XXM) H9N2 virus. The higher NA inhibition (NI) ability of N2-specific MAbs was found to result in greater proportions of mutations in HA head. The HA mutations affected the thermal stability, switched the binding preferences from α 2,6-linked sialic acid receptor to α 2,3-linked sialic acid receptor, and promoted viral growth in mice lungs. These mutations caused significant HA antigenic drift as they decreased hemagglutination inhibition (HI) titers. The evolutionary analysis also proved that some HA mutations were highly correlated with NA antibody pressure. Our data demonstrate that HA mutations caused by NA-specific antibodies affect HA properties and may have contributed to HA evolution.”

The first paragraph in the Results section also has been revised as following.

“MAbs A2A3, A4C6, A5D12, A3C9, A6A7 and B4D6 against NA of A/Chicken/Jiangsu/XXM/1999 (XXM) virus were previously proved to have neutralizing ability in micro-neutralization (MN) assay and high NA inhibition (NI) ability in enzyme-linked lectin assay (ELLA) (21). In this research, the NI ability of these MAbs to the wild-type (WT) virus was further measured in 2’-(4-Methylumbelliferyl)- α -D-N-acetylneuraminic acid (Mu-NANA) assay. In contrast, only MAbs A2A3, A3C9 and B4D6 have significant NI effect in Mu-NANA assay, while the other MAbs and a non-neutralizing MAb B6G5 all have very weak (<20%) or no NI effect on XXM virus (Figure 1A). The Mu-NANA assay result indicates that the MAbs A2A3, A3C9 and B4D6 can effectively block the Mu-NANA, which is a small molecular substrate, binding with the NA.”

3. The authors have made multiple statements without any evidence or supporting data, which need to be corrected or modified to improve the manuscript. Here are some examples.

1) On pages 9-10 lines 196-199, the authors stated that "In the 1990s and 2000s, the HA mutations as V198A, Q234L and the other antigenic sites in HA head were mainly caused by host adaptation and HA-specific antibodies induced by the inactivated H9N2 vaccines which were firstly used in China poultry in 1990s (Figure 5)." How do you know these mutations were mainly caused by host adaptation and vaccination?

Response : Thanks for your question. According to another research on effect of the selection pressure of vaccine antibodies (DOI:10.1186/s13568-020-01036-0), positions 198 and 234 post different mutant directions under the selection pressure. The A198 and Q234 in HA of WT A/Chicken/Shanghai/F/98(H9N2) virus in chicken passages became T198 and Q234 with antibody pressure but V198 and L234 without antibody pressure. Positions 198 and 234 (190 and 226 in H3 numbering) are both critical residues for receptor binding and antigenic structure (DOI: 10.1128/JVI.01141-16, DOI: 10.1016/j.vetmic.2010.05.010). A198 and L234 became dominant residues in HA of H9N2 which must be the result of host adaptation and vaccination.

2) Page 10 lines 208-210, the authors stated that "It should also be noted that the increase of human H9N2 infections in China in recent years may correlate with these HA mutations caused by NA antibody pressure." In fact the human infection with H9N2 virus is still rare and there is not obvious increase in the human cases, and more importantly, with the switch in the receptor preference of the mutants from alpha 2, 6 to alpha 2, 3, I would argue that these viruses would have a decreased ability to infect humans.

Response: Thanks for your question. In most cases, the infections of H9N2 in human are with very mild symptoms which are easily overlooked (DOI: 10.1016/j.cmi.2017.10.026). While the seroprevalence study showed high positive rates in poultry workers and other exposed populations (DOI: 10.3201/eid2512.190261, DOI: 10.3201/eid2407.172059). The receptor binding preferences affect the transmissibility of virus in different hosts. While the H9N2 virus in China poultry, which show high preferences for alpha 2, 6 receptor, can still well infect the chickens (DOI:10.1371/journal.ppat.1004508). Moreover, some virus like H7N9 virus with high preferences for alpha 2, 3 receptor poses higher pathogenicity to mice than other H7N9 viruses (DOI:10.1371/journal.ppat.1009561). In view of the possible misunderstanding, we deleted this sentence in revised manuscript.

3) Page 11 lines 239-240: "The HA mutations identified in this study also decrease the binding ability of serum samples of challenged mice..." Did you measure the binding ability?

Response: Thanks for your question. We are sorry for our inaccurate description. We have revised as following:

"The HA mutations identified in this study also decreased the HI titers of serum samples of challenged mice, H9-specific MAbs and chicken serum, which indicate that HA mutations caused by NA antibody pressure can also help virus evade from neutralizing antibodies against HA."

Minor comments

1. Page 5 lines 103-105: "The other MAbs A4C6, A5D12, B4D6 and A6A7 showed weaker NI activity in Mu-NANA assay, while still resulted in slight codon changes at same positions." What do you mean by "slight codon changes"? Did these also occur in HA? If yes, please specify these codon changes and the amino acid substitutions.

Response: Thank you for your nice question. As shown in line 108, all the HA mutations at 166, 198 and 234 were fixed. "The nucleotide codon of the position 166 changed from GAC to AAC, which resulted in a D166N mutation in HA (Figure 1B). Similarly, the dual nucleotide changes of ACA to GTA led to T198V mutation and another codon change of CTG to CAG led to L234Q mutation." We further marked the amino acid substitutions in new Figure 1B.

2. Pages 5-6 lines 110-111: "Moreover, HA mutations were also identified in 4 of 10 escape mutants purified by plaque assay before (Table 1)..." Were data displayed in Table 1 from the current study or a study published previously?

Response: Thank you for your nice question. The previous work only focused on NA mutations. The HA and NA mutations showed in Table 1 belong to the same mutants prepared in previous work. HA mutations were not published in previous work. In this research, the HA properties of these mutants were examined.

3. Page 7 lines 149-150: "All results show that inhibition of NA activity caused by NA antibody pressure would break the HA-NA balance (Figure 3)." There are no data to support that the HA-NA was off balance. How do you define the HA-NA balance? Do these mutants have enhanced/decreased NA activity or hemagglutination ability?

Response: Thank you for your nice question. The HA and NA can normally function, which forms the HA-NA balance. Here, the inhibition of NA by NA-specific MAbs led to the dysfunction of NA. HA can work while NA could not work, which resulted in the imbalance. The hemagglutination ability of the mutants was affected because the receptor binding preferences have switched, but all mutant viruses can still reach very high HA titers in SPF chicken embryos as WT virus. The NA ability of the mutants also had no significant changes according to our previous work (DOI: 10.1080/22221751.2021.1879602). The enhanced/decreased NA activity or hemagglutination ability need to be further tested in more specific and sensitive assays.

4. Page 8 lines 114-115: "however, all the mutants were detected at higher titers although not significant in mouse lungs at 3 days post-infection (Figure 4C)." What do you mean by "not significant"? Compared with what?

Response: Thanks for your question. The viral titers of mutant viruses challenged mice was higher than the WT group but with no significant differences. For better understanding, we have modified as following:

"However, all the mutant groups were detected higher but with no significant differences in viral titers of mice lungs than that in the WT group at 3 days post-infection (Figure 4C)"

5. Pages 9-10 lines 182-210. While the authors were emphasizing the correlation between the HA mutations with NA antibodies, please do note that mutations at H9 positions 168 and 201 can also be selected by HA-specific antibodies as the authors' group has reported previously (Wan et al.,

JVI, 2014).

Response: Thanks for your question. Most residues like 166, 168, 198, 201 were also involved in binding HA-specific antibodies. These residues are also very flexible and able to help viruses escape from HA-specific antibodies by substitutions. In this study, we identified HA mutations at these positions in HA cause by NA antibodies which have been proved to have no HI ability. Therefore, we concluded that these HA mutations were caused by NA-specific antibodies. NA-specific antibodies resulted in dysfunction of NA which broke the HA-NA balance. This imbalance led to not only the mutations in NA but also the substitutions of flexible residues like 166, 168, 198, 201 in HA to help virus survive from NA antibody pressure and infect new cells as soon as possible.

6. Page 14 lines 296-301. Please provide details about the western blot analysis, including the amount of each virus loaded, the treatment and conditions (with or without reductant, the buffer used, etc.) for the SDS-PAGE, the transfer of the viral proteins to the film, and the antibody used for probing the target and the concentration of the antibody, etc.

Response: Thanks for your advice. We have modified the description as following:

“Allantoic fluid of each virus was used for non-reducing polyacrylamide gel electrophoresis as previously described (48). The allantoic fluids were treated with loading buffer without DL-dithiothreitol (DTT). Treated samples were used for sodium dodecyl sulfate polyacrylamide gel electrophoresis (SDS-PAGE) and then transferred to nitrocellulose membranes (GE, Massachusetts, USA) for Western blot analysis. The H9-specific 2G4 MAb was diluted in 5000 folds with PBS containing 0.5% tween-20 (PBST) and used as primary antibody for HA protein determination. After washing six times with PBST, peroxidase conjugated goat anti-mouse IgG (H+L) antibody (Jackson ImmunoResearch, Pennsylvania, USA) was diluted in 10000 folds and used as secondary antibody. After washing another six times with PBST, membrane was finally immersed in Immun-Star HRP Substrate (Bio-rad, California, USA), and the chemiluminescent signals were observed with a FluorChemE imaging system (Protein Simple, California, USA).”

7. Page 16 lines 350-351: "...on the days 4 and 6 post-infection, and the lungs were collected for virus titration. The lung tissues were collected at day 6 post-infection and fixed with 4% formaldehyde for histopathological examination." Please specify how you have managed to divide each of the day 6 lung samples for viral titration and histopathological examination.

Response: Thanks for your question. In mouse experiment of the day 6, each mouse lung was divided into left lung and the right lung. The left lung of the mouse was fixed with 4% formaldehyde for histopathological examination and the right lung of the mouse was added into 0.5mL Opti-MEM with 2 µg/mL TPCK trypsin for virus titration.

8. Please change "a" and "b" in the receptor analogs to α (alpha) and β (beta), wherever appropriate.

Response: Thanks for your advice. We have modified them in the revised manuscript.

9. The authors are wished to be cautious in stating throughout the text that the HA mutations were caused by NA antibodies or NA antibody pressure. This may give the readers the impression that NA antibodies directly interacted with HA in your assay. I believe that the HA mutations were

introduced to adapt to the NA mutations.

Response: Thanks for your question. We have detected HA mutations in viruses selected by MAb A3C9 before detection of NA mutations. Moreover, larger proportions of mutations in HA than in NA can be detected in viruses selected by MAb A2A3. For viruses selected by other MAbs, HA mutations and NA mutations took place at the same time. Therefore, we prefer to believe HA mutations are coming from NA antibody pressure rather than compensatory mutations for NA mutations. NA mutations here did not significantly affect the NA enzyme activity and viral growth of other mutants without HA mutations. Therefore, we cannot conclude that HA mutations were introduced to adapt to the NA mutations.

10. Tables 1-3, please provide necessary information including the viruses used in your study (for instance, what does "WT" mean?) and the source of antibodies/sera for your analyses.

Response: Thanks for your advice. The "WT" is abbreviation for "wild -type", we have added the detailed information in Results part at line 92. MAbs against HA of H9N2 AIV were prepared in reference 37. The source of sera has been provided in Materials and method part at line 357.

11. Figure 3. Please explain more in the figure legend so that the readers can understand what message you want to convey through the schematic representation.

Response: Thanks for your advice. The graph has been modified. More information has been provided in legend as following:

"The sialic acid receptors are shown in the form of blue diamond. The blue see-saws are used to show HA-NA function balance. The inhibition of NA by NA-specific antibodies results in dysfunction of NA which further lead to the imbalance of HA-NA function. When the pressure transmits to the flexible HA, HA mutations take place in HA head and change receptor binding preferences for survival under NA antibody pressure."

November 30, 2021

Dr. Aijian Qin
Yangzhou University
College of Veterinary Medicine
12 East wenhui road
Yangzhou 225009
China

Re: Spectrum01439-21R1 (Identification of haemagglutinin mutations caused by neuraminidase antibody pressure)

Dear Dr. Aijian Qin:

Your manuscript has been accepted, and I am forwarding it to the ASM Journals Department for publication. You will be notified when your proofs are ready to be viewed.

Sincerely,

Heba Mostafa
Editor, Microbiology Spectrum
